# Privatization and Multi-Fatality Disasters: A Causal Connection Exposing Both Worker and Citizen Health and Safety Failures in the UK?

**DOI:** 10.3390/ijerph192013138

**Published:** 2022-10-12

**Authors:** Matthias Beck, Andrew Watterson

**Affiliations:** 1Department of Management & Marketing, University College Cork, National University of Ireland, T12 K8AF Cork, Ireland; 2Faculty of Health Sciences and Sport, Stirling University, Scotland FK9 4LA, UK

**Keywords:** technological disasters, privatization, multi-fatalities, deregulation

## Abstract

Although several countries have experienced large-scale privatization initiatives, relatively little is known about the impact of these initiatives on the health and safety of workers and resident populations. Examining data on technical (as compared to natural) multi-fatality disasters collected in the WHO’s Emergency Events Database (EM-DAT) for the UK and a number of European comparator countries for recent decades, this paper shows that the incidence of these disasters and the number of deaths resulting from them rose significantly in the UK during the period from 1979 to 1991 when the country engaged in extensive and aggressive privatization campaigns which were supported by several consecutive Conservative governments. This observed UK blip or abnormal increase in multi-fatality disasters is apparent for the UK both in terms of a “within-country” comparison (i.e., when we compare the privatization period of 1979 to 1997 with other periods), as well as when viewed in terms of comparisons with the comparable European countries of Germany, France, and Italy for the same period (1979 to 1997). Contrary to previous analyses which suggested that there is no clear link between privatization and deterioration of health and safety, this paper concludes that the UK privatization experience (1979–1997) provides robust country-specific evidence of privatization initiatives leading to increases in the number of multi-fatality technological disasters as well as related fatalities. This evidence should be seen as a deterrent to similarly extensive and aggressive initiatives which, particularly in less developed countries, could result in similarly disastrous outcomes.

## 1. Introduction

The election of Margaret Thatcher as Prime Minister in 1979 set in motion a period of wide-spread privatizations which included public utilities and formerly nationalized industries in the communications, energy, and transport sectors [1]. This privatization wave continued under John Major and lasted until 1997 when Labour returned to power. Overall these measures were accompanied by deregulatory policies, which sought to reduce state oversight and control and shrink the size of the public sector workforce in general. Subsequent Labour Governments made some efforts to reverse these measures, while continuing to support a more nuanced approach to private sector involvement in public services provision, so that the period from 1979 to 1997 can be seen as a distinct and discrete phase during which the political economy of the UK was restructured in fundamental ways.

Following this experience of privatization in the UK, there has now been a period of reflection during which the socio-economic consequences of these activities have been examined by researchers and advocacy groups. In as far as the impact of privatization on accident rates and disaster-proneness are concerned, there appear to be industry-specific evidence of adverse effects. This includes the impact of privatization on rail safety in the UK [2]. Some aspects of these analyses have remained somewhat inconclusive. Thus, it remains unclear whether privatization by itself is risky on account of changes or disruptions of governance arrangement, or whether only those forms of privatization which are associated with right wing pushes from segments of capital seeking to advance privatization and deregulation are prone to erode safety. While many of these investigations have centered on the economic effects of these initiatives, far less has been written about their impact on health and safety. One exception to this is an oft-cited review paper, which appeared in the Journal of Epidemiological and Community Health in 2007. Examining a total of eleven “highly heterogeneous” papers, including ten UK based and one Portuguese study, which evaluated the “health impacts of privatization of building, water, paper, cement, bus, rail, mining, electric, and gas companies” this paper concluded that “no robust evidence was found to link privatization with increased injury rates to employees and customers” [3].

Although this meta-analysis sought to provide a balanced and systematic review of the literature accessed by the authors, this conclusion is problematic for several reasons. Firstly, and perhaps most importantly, the underlying studies examined by the authors present individual industries which do not necessarily mirror the full range of sectors which were affected by privatization. Secondly, of the eight UK based studies investigating the impact of privatization of health and safety, only three include statistical tests of any kind. While it is self-evident that meta-analyses of this kind have to rely on the available literature, it can be argued that these issues could alternatively, and perhaps more fruitfully, be examined on the basis of aggregate statistical data on fatalities or serious injuries or on the basis of relevant data on the incidence of disasters. In fact, it is interesting to note that the authors conclude that “given the media and political interest in rail safety following the privatization of the British Rail, we were particularly surprised by the lack of more specific and comprehensive evidence on injuries to employees and the public” [3].

During the period from 1979 to 1997, the UK did indeed experience an unusually large number of multi-fatality disasters including many such disasters in the rail transport sectors more generally. UK rail disasters of this period alone comprise, among others: in 1987, the Forest Gate collision (14 injured) and the Towy bridge incident (4 fatalities); in 1988, the St Helens, Merseyside commuter train incident (causing the death of the driver and injuries to 18 passengers), the Clapham junction collision (35 fatalities), the Glasgow train collision (2 fatalities), the Newcastle Central Station collision (15 injured); in 1990 the Stafford Station crash (which killed the driver and injured 35), the London Cannon Street incident (injuring 240); in 1991, the River Seven Tunnel collision (injuring over 100); in 1994, the Cowden crash (5 fatalities and 12 injuries); in 1995, the Aisgill accident (killing a guard and injuring 30); in 1996, the Watford Junction collision (with 1 fatality and 69 injuries); and in 1997, the Southall crash (with 7 fatalities). Other notable disasters of the period, include maritime disasters (involving private rather than privatized companies) such as the capsizing of the British ferry Herald of Free Enterprise (with 193 fatalities) in 1987; the sinking of the Marchioness pleasure boat (with 51 fatalities) in 1989; the explosion of the offshore platform Piper Alpha (167 fatalities) in 1988; and the 1993 offshore helicopter crash near the Cormorant Alpha platform (11 fatalities).

Although not all major disasters which occurred during this period can be directly linked to privatization and deregulation measures, it has been convincingly argued that an atmosphere of privatization and deregulation can affect safety on several levels. Thus, a study of the Exxon Valdez oil spill noted that this incident was aggravated and potentially triggered by the interplay of a number of factors, including Exxon’s cutbacks in tanker staff and oil spill experts, personnel reductions by the Coast Guard, as well as the absence of escort ships and high power radar [4]. In other words, there are reasons to assume that privatization and deregulation initiatives can affect the safety performance of safety-critical companies or activities on a number of closely related tangible and less tangible levels, where multiple factors can interact and exacerbate each other.

### 1.1. Privatization

Privatization describes a number of activities which are often carried out by government agencies in bundled form [5]. At the most basic level, privatization involves the sale, or partial sale, of state-owned enterprises to the private sector. A less “intense” form of privatization involves processes where state agencies contract with the private sector for the provision of certain formerly public services, while they remain in overall control. Similarly, the term privatization is sometimes applied to circumstance where public sector agencies are forced to operate under market conditions, for instance, by raising capital from the private sector or selling services under market conditions in competition with private companies [6]. Another use of the term privatization describes public-private partnership arrangements which usually involve the collaboration of a government agency with private financiers and contractors in the procurement of infrastructure or services. Lastly, and perhaps misleadingly, privatization has also been applied to the removal of existing state control or state regulatory requirements, such as licensing requirements or health and safety rules, from certain industries or activities.

Characteristically, most governments which embark on privatization initiatives engage in bundles of measures which include several of the different activities listed above. As a consequence, large-scale privatization drives can impact broadly on national and/or regional economies, creating changes which simultaneously affect workers, clientele, communities as well as managerial and regulatory practices. These changes, moreover, often reach beyond the sectors or industries which are directly targeted by privatization initiatives, affecting related sectors, as well as seemingly separate spheres of activity where the new models of privatization are emulated.

### 1.2. Privatisation in the UK

The period of privatization which the UK underwent during 1980s and 1990s can be described as extraordinarily intense on account of a number of factors, including its duration, extent and the underlying ideological zeal with which it was pursued [7]. Firstly, as regards duration, privatization remained one of the core components of Conservative domestic policy over a period of five consecutive Governments; three under Margaret Thatcher from 1979 to 1990 and two under John Major from 1990 to 1997. Secondly, following initial announcements of the intent to privatize nationalized industries, the UK privatization agenda was gradually broadened to include, among others, the compulsory contracting out of direct labour services in local government and health. Thirdly, it has been convincingly argued that throughout this period the Conservatives’ ideological commitment to privatization was so central that it often overrode other economic rationales such as the promotion of competition or the increase of public revenue.

Overall, this period of privatization has been described as having encompassed seven major components. These included, firstly special asset sales which incorporated the denationalization of British Gas, British Airways and British Telecom as well as the sale of public sector companies previously acquired by the state, such as British Petroleum. Secondly, it involved the deregulation of monopolies which exposed public sector companies to competition, as in the bus sector. Thirdly, it involved the compulsory contracting out of labor services which had been previously directly performed by local government, the National Health Service and the Civil Service. Fourthly, it actively encouraged the private sector to engage in the provision of what had been public services in the form of private care homes for elderly and other vulnerable groups. Fifthly, it involved the creation of special units within the public sector which were tasked with adopting private sector or commercial practices in areas such as the redevelopment of deprived localities. Sixthly, there was a strong focus on the reduction of subsidies and the introduction of user charges particularly with regard to welfare services. Lastly, the Conservatives aggressively pursued the sale of subsidized and publicly owned housing, thereby drastically reducing the available stock of social housing [8].

It has been documented that by 1991, over 50% of the public sector had been transferred to the private sector, with 650,000 workers changing sectors [9]. Additionally by that time, more than 1,250,000 publicly owned houses had been sold, while contracting out arrangements had replaced most direct labor services [9]. These activities, combined with a decline in UK manufacturing, impacted severely on the balance of power between workers and employees via a number of factors. Firstly, contracting out in the public sector created significant job losses which further aggravated the tight labour market situation of the time. Secondly, the loss of these jobs, or their transfer to private sector companies, undermined gains trade unions had made in the public sector. This will have impacted on UK occupational health and safety on account of the fact that UK trade union safety representatives had rights to inspect and access to information that helped to ensure better reporting and recording of injuries and hazardous incidents. Thirdly, as far as transferred workers were concerned, there was a measurable deterioration of working conditions, especially with regard to holidays and overtime [9]. Lastly, while the Conservatives withheld a direct de-regulatory offensive on occupational safety, there was a gradual continual undermining of the ability of regulatory agencies at the local and national level to fulfil their functions in this area [10].

## 2. Materials and Methods

The following analysis is based on data on technological, as compared to natural, disasters which were listed on the Emergency Events Database (EM-DAT) for the period from 1970 to 2007. EM-DAT, which has been created with the initial support of the WHO and the Belgian Government, is currently hosted by the University of Louvain. As one of the largest disaster databases in the world, this publicly accessible resource includes essential data on the occurrence and effects of over 12,800 mass disasters from 1900 to the present. The database itself has been compiled from a number of sources including UN agencies, non-governmental organizations, insurance companies, research institutes and press agencies [11]. For a disaster to be included in the database it has to meet one of the following criteria: (i) 10 or more people being reported as killed, (ii) 100 or more people being reported as affected, (iii) a declaration of a state of emergency being issued, or (iv) a call for international assistance being issued [11].

In order to avoid ambiguities, the consecutive analysis is limited to only those technical disasters which resulted in 10 or more people being reported as killed. Data on these disasters were retrieved for the countries: UK, Germany, France, and Italy, initially using the de-limiter “technical disasters”. The choice of these countries was based on their comparable level of development, their location within the European region, as well as roughly comparable population sizes; with 1995 populations of 58 million for the UK, 80.6 million for post-reunification Germany, 57.4 million for France and 57.8 million for Italy. For purposes of simplicity and accuracy figures for Germany refer to West Germany only up until reunification in 1990, and thereafter to the re-united Germany.

The tabular presentation of data is supplemented by the calculation of simple differences in Poisson Rates (difference between two measures of the number of cases/total person years, analogous to difference in incidence rates), presented as a footnote in each table. This difference test is used when count data for a sample are compared with a target. In this case, the difference between incidence rates during one time period compared to another time period, or in one country compared to another country (during the same time period). It should be noted that the test may have low power given the low number of occurrences in some cases and the relationship between the size of the intervals [12]. Ng and Tang (2005) recommend changing the respective time intervals (and/or log transforming) the statistics as a robustness test [13]. For the purpose of this analysis we conducted a robustness test where we combined the ‘non-privatization’ intervals of out tables (see below, Section 3.1) for 1998–2007 and 1970–1978 and contrasted this with the privatization interval of 1979–1997. The same procedure was performed forall Table A2, Table A3 and Table A4. These robustness checks are presented in Appendix A.

In order to identify a potential effects of privatization on the incidence of technical disasters, the UK time series for 1970 to 2007 was divided into three segments. For the UK this included, (i) a pre-privatization period from 1970 to 1978, (ii) a privatization phase from 1979 to 1997, which coincides with the period during which the Conservatives held the government, and (iii) a current period from 1998 to 2007. To facilitate comparisons across countries, the same segmentation was applied to the other three countries (Germany, France and Italy), which, while being governed by different political parties over time, did not experience comparable privatization drives.

## 3. Results and Discussion

The following analytical section comprises four subsections. The first subsection compares the incidence of multi-fatality technical disasters defined as disasters resulting in 10 or more deaths for the aforementioned time periods (Table 1). The next subsection examines the total death count resulting from these disasters based on the same data segments (Table 2). The subsequent part of the data analysis then re-examines the same variables with air accidents having been removed from the dataset. The reason for this is that air accidents can involve foreign carriers which are not necessarily affected by the regulatory climate of individual countries. Again these data are reported for the number of disasters which occurred (Table 3) as well as their aggregate death toll (Table 4). In Table 1 we find the difference in Piosson rate to be moderately significant (0.19 level, one tailed) for the comparison of the number of UK multi-fatality disasters (of 10 or more fatalities) occurring during the Conservative governed privatization period from 1979–1997 (27 total), to the number occurring from 1998 to 2007 (4 total) when Labour was back in power.

### 3.1. Comparison of the Incidence of Multi-Fatality Technical Disasters Defined as Disasters Resulting in 10 or More Deaths 1970–2007

Table 1 shows the total number of all multi-fatality technical disasters (resulting in 10 or more deaths for each of the time periods (sub-labelled a to c) for the four countries (sub-labelled 1 to 4) together with the number of such disasters per year.

**Table 1 ijerph-19-13138-t001:** Total and per Year Number of Multi-fatality Disasters, 1970–2007.

**Country**	**UK(1)**	**Germany (2)**	**France (3)**	**Italy (4)**
**Time Period**				
1998–2007 (a)	4	7	18	15
*Per year*	*0.40*	*0.70*	*1.80*	*1.50*
1979–1997 (b)	27	12	26	21
*Per year*	*1.42*	0.63	1.37	1.10
1970–1978 (c)	10	5	10	2
*Per year*	*1.11*	*0.55*	*1.10*	*0.22*
**Comparison of Poisson Rates**				
Within Country (UK):		(1b) to (1c) *p* = 0.624		
		(1b) to (1a) *p* = 0.019 *		
Across Countries for 1979–1997		(1b) to (2b) *p* = 0.025		
		(1b) to (3b) *p* = 1.000		
		(1b) to (4b) *p* = 0.470		

* significant at 0.02 level.

In terms of aggregate figures, the UK figure of 27 multi-fatality disasters during the privatization period (1979–1997), is the highest for any country and any time period. While this seems to lend support to a hypothesis that there was a general deterioration of health and safety during this period, this finding is not necessarily confirmed by examining the “per year” rate for the incidence of disasters across all countries and time periods. Thus, the admittedly very high UK rate of 1.42 disasters per year for the period from 1979–1997 is eclipsed by the rate for France for the most recent period from 1998–2007 (1.80) and, to a lesser degree, by that for Italy for the same period (1.50). Statistically also, a potentially significant result can only be detected for a UK within-country comparison of the 1979–1990 period with the most recent 1998–2007 period (at the 0.02 level).

### 3.2. The Total Death Count Resulting from These Disasters Based on the Same Data Segments

Following on from the previous table, Table 2 shows the total number of fatalities which resulted from all of these multi-fatality technical disasters; again for each of the time periods (sub-labelled a to c) for the four countries (sub-labelled 1 to 4) together with the number of fatalities per year.

**Table 2 ijerph-19-13138-t002:** Total and per Year Number of Fatalities from Multi-fatality Disasters, 1970–2007.

**Country**	**UK(1)**	**Germany (2)**	**France (3)**	**Italy (4)**
**Time Period**				
1998–2007 (a)	69	244	406	477
*Per year*	*6.9*	*24.4*	*40.6*	*47.7*
1979–1997 (b)	943	254	790	744
*Per year*	*49.6*	*13.3*	*41.5*	*39.1*
1970–1978 (c)	401	164	1000	85
*Per year*	*44.5*	*18.2*	*111.1*	*9.4*
**Comparison of Poisson Rates**				
Within Country (UK):		(1b) to (1c) *p* = 0.074		
		(1b) to (1a) *p* = 0.000 **		
Across Countries for 1979–1997		(1b) to (2b) *p* = 0.000 **		
		(1b) to (3b) *p* = 0.000 **		
		(1b) to (4b) *p* = 0.000 **		

** significant at 0.001 level.

In terms of aggregate figures, the UK figure of 943 deaths from multi-fatality disasters during the privatization period (1979–1997), is the second highest for any country, exceeded only by that of France for the earlier period from 1970–1978. This initial support for the hypothesis that there was a general deterioration of health and safety during this period which resulted in an increased number of fatalities is confirmed by examining the “per year” rate for the incidence of disaster-related deaths across all countries and time periods. Thus, the UK rate of 49.6 disaster-related fatalities per year for the period from 1979–1997 is again only exceeded by the rate for France for the earlier period from 1970–1978 (111.1). Statistically the Poisson Rate difference test provides strong support for the presence of an abnormally high number of disaster-related fatalities in the UK during the privatization period (1979–1997), both in terms of within-country (1b = 243 compared to 1a = 69) and across-country comparisons (1b [UK], 1979–1997 = 943 to 2b [Germany], 1979–1997 = 254); (1b [UK], 1979–1997 = 943 to 3b [France], 1979–1997 = 790) and (1b [UK], 1979–1997 = 943 to 4b [Italy], 1979–1997 = 744). Thus, the hypothesized UK imbalance is evidenced in within-country comparisons by a statistically significant *p* value for a comparison of the 1979–1997 period with the most recent period (1998–2007) and perhaps more importantly, all across-country for the critical period (1979–1997) which show a statistically significant elevation for the UK.

### 3.3. The Number of Disasters That Occurred between 1998–2007

In order to remove potential distortions which may have resulted from the inclusion of air disasters in the data set, these have been removed for the second part of the analysis. Again, the resultant data are first examined on the basis of an aggregate count of multi-fatality technical disasters (see Table 3).

**Table 3 ijerph-19-13138-t003:** Total and per Year Number of Multi-fatality Disasters (Excluding Air), 1970–2007.

**Country**	**UK(1)**	**Germany (2)**	**France (3)**	**Italy (4)**
**Time Period**				
1998–2007 (a)	3	5	16	13
*Per year*	*0.30*	*0.50*	*1.60*	*1.30*
1979–1997 (b)	22	11	17	17
*Per year*	*1.15*	*0.58*	*0.89*	*0.89*
1970–1978 (c)	8	5	6	2
*Per year*	*0.89*	*0.55*	*0.67*	*0.22*
**Comparison of Poisson Rates**				
Within Country (UK):		(1b) to (1c) *p* = 0.655		
		(1b) to (1a) *p* = 0.031		
Across Countries for 1979–1997		(1b) to (2b) *p* = 0.081		
		(1b) to (3b) *p* = 0.522		
		(1b) to (4b) *p* = 0.522		

The removal of air accidents from the dataset shown in Table 3 changes the findings of the analysis regarding the number of multi-fatality disasters very little (compare to Table 1). The UK continues to have the highest figure of these disasters for any country and any time period for the period from 1979–1997 with 22 disasters. However, the statistical significance of this is not confirmed by looking at per year disaster figures or the corresponding statistical tests.

### 3.4. Aggregate Death Toll of the Disasters between 1998–2007

Following on from the previous format, Table 4 shows the total number of fatalities which resulted from all of these multi-fatality technical disasters excluding air disasters; again for each of the time periods (sub-labelled a to c) for the four countries (sub-labelled 1 to 4) together with the number of fatalities per year.

**Table 4 ijerph-19-13138-t004:** Total and per Year Number of Fatalities from Multi-fatality Disasters (Excluding Air Accidents), 1970–2007.

**Country**	**UK(1)**	**Germany (2)**	**France (3)**	**Italy (4)**
**Time Period**				
1998–2007 (a)	58	162	269	343
*Per year*	*5.8*	*16.2*	*26.9*	*34.3*
1979–1997 (b)	763	242	442	614
*Per year*	*40.1*	*12.7*	*23.3*	*32.3*
1970–1978 (c)	283	164	403	85
*Per year*	*31.4*	*18.2*	*44.7*	*9.4*
**Comparison of Poisson Rates**				
Within Country (UK):		(1b) to (1c) *p* = 0.000 **		
		(1b) to (1a) *p* = 0.000 **		
Across Countries for 1979–1997		(1b) to (2b) *p* = 0.000 **		
		(1b) to (3b) *p* = 0.000 **		
		(1b) to (4b) *p* = 0.000 **		

** significant at 0.001 level.

Having excluded air accidents from the dataset, the UK figure of 763 deaths from multi-fatality disasters during the privatization period (1979–1997), is now the highest for any country and for any time period; lending strong support for the hypothesis that there was an increase in disaster-related deaths during the privatization period of the UK. This analysis is confirmed by a visual inspection of the per year data as well as, more importantly all within-country and across-country statistical tests for the critical period (1979–1997).

## 4. Conclusions

During its period of aggressive privatization, the UK has seen an increase in the number of multi-fatality disasters, and, perhaps more importantly a statistically significant abnormality in the number of deaths resulting from these disasters. This pattern can be evidenced both in terms of a comparison with UK data for earlier and later years, as well as on the basis of comparisons with other large European countries.

Tony Blair after 1997 also supported some aspects of privatization [14] albeit with less intensity, continuing some of the twin track of overt and covert deregulatory policies on occupational health and safety begun in the 1980s [15]. Some of these deregulatory policies effectively facilitated privatization schemes while their adverse effects on workers and the public were kept from close critical scrutiny [16]. However, in terms of a combination of intense outright privatization and deregulation, the period of Conservative rule from 1979 to 1997 must be seen as unique and in this sense it should perhaps also not surprise us that it that this period saw a statistically significant increase or blip in multi-fatality disasters which we document here. Overall there appears to be a need for the link between privatization, various political agendas and multifatalty disasters to be investigated more fully. In as far as privatization initiatives in developing and middle income economies are concerned, past research points to the specific vulnerability of these regions. Accordingly, it has been argued that social, political and economic structures in developing countries make them more vulnerable to disasters in general, and also to the potential adverse effects of privatization and deregulation [17]. Additionally, it has been argued that where such disasters occur, they tend to have more catastrophic consequences than elsewhere [18], with high density urban areas being particularly exposed [19].

## Data Availability

Raw data for the study can be found at https://emdat.be/. For details on data preparation and statistical analysis contact the corresponding author.

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
