# Peer review of "Privatization and Multi-Fatality Disasters: A Causal Connection Exposing Both Worker and Citizen Health and Safety Failures in the UK?"

_ijerph, 2022, doi:10.3390/ijerph192013138_

Round 1
Reviewer 1 Report
This paper needs a discussion section and stronger statements in the Conclusion. In the abstract you note that you present this analysis as a caution for the application of privatization in industrially/economically developing nations. You don't discuss that at all in the paper, and though of course you could eliminate that statement from the abstract, it really is an important point for your work.
As to discussion - you hardly discuss or provide references to explain why the number of disasters and related fatalities dropped during the Labor government period? And what is the reader to conclude seeing that the number of fatalities and disasters during the Blair period is lower than the base period of greater public ownership? You note that privatization continued during the Labor government period - so is privatization not so dangerous and instead the problem is the Conservatives' approach to it? Your discussion should address this. Is your paper just a warning against implementing privatization without regulation of facility safety and worker health and safety, or is it pointing to the deterioration of worker H&S under privatization of public operations?
What do you say to those countries undergoing early implementation of privatization, or countries like the UK and US where deeply right wing pushes from segments of capital seek to advance privatization and deregulation in order to weaken or eliminate the aspects of national governments that have been in place post-WWII to protect the working class from the ravages of capitalism?
At the very least note that your analysis points to the need for deeper research into how privatization of public operations is impacting worker health and safety.

Reviewer 2 Report
This paper explores the interesting question of whether privatization as carried out in the United Kingdom under the Conservative governments of the 1980s and '90s increased the number and severity of multi-fatality disasters. The authors claim that this is an under-explored area of research, which does seem to be the case. Therefore, the paper has the potential to make an important contribution to the field.
The paper suffers from two primary weaknesses, though each of these has the potential to be addressed and improved through revisions. First, the framing of the work needs stronger justification. While, as noted above, there does appear to be a dearth of literature on the relationship between privatization and multi-fatality disasters, the authors should discuss a few closely related pieces of research that would bolster their claims. For example, de Jong et al.'s 2016 paper "The impact of restructuring on employee well-being: a systematic review of longitudinal studies" in Work & Stress, 30(1): 91-114 examines work on restructuring more broadly (including privatization) and finds significant health impacts on employees, which may be used to support the idea that privatized organizations with less satisfied employees may find it more difficult to maintain a culture of safety. Falkenberg et al.'s 2009 paper "How are employees at different levels affected by privatization? A longitudinal study of two Swedish hospitals," in the Journal of Occupational and Organizational Psychology, 82: 45-65 comes to similar conclusions. More firmly establishing the links between the kinds of restructuring that privatized firms in the UK underwent and the conditions that might lead to higher rates of disasters would make the paper more convincing.
Additionally, the methodology of the paper is somewhat problematic. Removing air disasters makes sense as the authors note, but the emphasis is placed on the aggregate results presented first. When air disasters are excluded (Table 3), the Comparison of Poisson Rates is insignificant.
However, the use of such a statistical test may not be appropriate. One key assumption of a Poisson model is the independence of each occurrence from the others. That is, the occurrence of one disaster should not affect the probability of future disasters. But because multi-fatality disasters draw scrutiny, investigations, and, often, changes in laws or company policies, such independence does not hold, at least within sectors.
Finally, even if the assumption of independence is granted, the test may have very low power given the low number of occurrences in many cases and the relationship between the size of the intervals. Ng and Tang's 2005 paper "Testing the equality of two Poisson means using the rate ratio," in Statistics in Medicine 24: 955-965, for instance, finds this to be the case. They suggest changing the respective time intervals and log transforming the statistics to alleviate this issue. Logically, it also seems problematic that the main period in question is 18 years long, while the pre-privatization period is only 8 years and the post-privatization period is 9. Given the scope of the database used, extending the periods seems reasonable - though it is understood that there is a tradeoff between doing this and attempting to have each period represent an ideological shift, which may not lend itself to segments of equal length.
Other minor corrections needed:
-- On line 44 (p. 1), there is a small typo. The sentence should read "far less has been written" instead of "far less is has been written."
-- On line 221 (p. 5), the number for France in Table 1 is attributed to Germany
Round 2
Reviewer 2 Report
The authors' engagement with previous comments is welcome and sees the paper improved, with a clearer framing and firmer foundation in the literature on privatization. The revised conclusion section is also a welcome improvement that enhances the broader relevance of the work and provides good suggestions for future research.
The authors include information suggested in the previous report in footnote 12, and this is a welcome addition. However, the issue of low statistical power mentioned previously remains. The authors note that they do not change the time intervals being examined (and this is understandable given the tradeoffs mentioned in the previous report and that the authors acknowledge). It is unclear whether they perform any kind of log transformation, though, since they mention that suggestion from Ng et al. (2005) and then do not address it.
If the authors can more substantively address the methodological issues, beyond an acknowledgement of it, the paper seems well worth publishing. They should consider log-transforming the data, some kind of robustness check in which the intervals being examined are changed and results compared to the default, and changing how the data are presented and discussed (reflecting comments on this in the previous report).
Author Response
see pdf
